# Low-Cost Sensory Glove for Human–Robot Collaboration in Advanced Manufacturing Systems †

**Tyrone Bright *** , **Sarp Adali** and **Glen Bright**

Department of Mechanical Engineering, University of KwaZulu-Natal,
Durban 4001, KwaZulu-Natal, South Africa; adali@ukzn.ac.za (S.A.); brightg@ukzn.ac.za (G.B.)

* Correspondence: tyronebright1997@gmail.com

† Adapted from: Bright, T.; Bright, G.; Adali, S. Close Human Robot Collaboration by means of a Low-cost Sensory Glove for Advanced Manufacturing. In Proceedings of the International Conference on Electrical, Computer, Communications and Mechatronics Engineering (ICECCME), Grand Baie, Mauritius, 7–8 October 2021.

**Abstract:** Human–robot collaboration (HRC) enables humans and robots to coexist in the same working environment by performing production operations together. HRC systems are used in advanced manufacturing to improve the productivity and efficiency of a manufacturing process. The question is which HRC systems can ensure that humans can work with robots in a safe environment. This present study proposes a solution through the development of a low-cost sensory glove. This glove was developed using a number of hardware and software tools. The sensory glove analysed and computed the motion and orientation of a worker's hand. This was carried out to operate the robot through commands and actions while under safe operating conditions. The sensory glove was built as a mechatronic device and was controlled by an algorithm that was designed and developed to compute the data and create a three-dimensional render of the glove as it moved. The image produced enabled the robot to recognize the worker's hand when collaboration began. Tests were conducted to determine the accuracy, dynamic range and practicality of the system. The results showed that the sensory glove is an innovative low-cost solution for humans and robots to collaborate safely. The sensory glove was able to provide a safe working environment for humans and robots to collaborate on operations together.

**Keywords:** human–robot collaboration; sensory glove; flexible manufacturing systems; industry 4.0

## 1. Introduction

A human–robot collaboration (HRC) environment involves humans and robots working together, in a safe environment, towards a common goal. The subject area has been reviewed in [1]. Safety issues involving human–robot collaboration have been an important subject for obvious reasons and have been investigated in a number of studies [2–4]. Collaborative robotics involving safety is a subject of importance for the factories of the future and the subject has been reviewed in [5,6].

Recent articles discuss several issues concerning safety in a human–robot collaboration environment and study the standards applicable in these settings [7,8] These issues are centered around the notion that it is not possible to ensure the complete safety of the worker in an HRC environment. Extensive research in HRC environments has been focused on camera-based solutions, such as augmented reality (AR) and virtual reality (VR) solutions in [9–11]. A recent review, conducted by [12], highlights the safety challenges that limit VR and AR in the HRC environment. This includes the limitations of camera technology in a dynamic environment. Manufacturing floors are dynamic environments that include a large amount of moving parts and unforeseen events. For example, light and dust can affect the object detection and processing of 2D cameras as this corrupts the image. AR and VR systems require a controlled and structured environment to perform at their

best. AR technology is limited due to the field of view and tends to create dizziness in workers according to [12]. VR technology is limited due to controller connections and the workspace as they require cable attachments to a computer. This all negatively affects the over performance of such systems and the overall safety of the worker.

In comparison, less research has been conducted on sensor-based approaches to the creation of HRC environments. It is believed that sensor-based approaches have the potential to provide greater safety in a dynamic and changing environment such as manufacturing. Before we explore sensor-based HRC systems, we need a broad ideology of human–robot collaborative systems.

The process of collaborating with a human was defined by [13] where they outlined the broad ideology of HRC systems. The ideology of HRC systems focused on three core principles. These core principles revolve around coexistence, collaboration and safety as discussed in [14,15]. These principles were established to create an efficient, productive and safe environment where workers could work with robots in creating a product or services as discussed in [16].

These principles of HRC are what made sensor-based technology, specifically wearable technology, a promising area of research for manufacturing, production or assembly systems. Wearable technology, in particular sensory gloves, has received increased research and development in the human–robot collaboration field. This is evident as various sensory gloves have been designed and developed over recent years as documented in [17,18].

Such gloves are mostly used in flexible manufacturing systems (FMS) and are designed to easily adapt and change to the type and quantity of a product [19]. Due to recent advances and developments, wearable technology has been shown to be a creative technology in this field as it enables humans to enhance the flexibility of the system. This solution eliminates the limitations of imagine processing, which is a weakness of camera-based technology, and increases the potential workspace, as they can be used wirelessly. These developments show the great potential for their use in an HRC environment. The development of these system has been a costly approach. Priority software and the advancements in chips have caused the development costs to rise. A sensory glove built by a commercial company known as Sensoryx retails at CHF 750 (approximately ZAR 12500).

Therefore, the objective of the present work was to design a low-cost mechatronic sensory glove to enable humans and robots to collaborate in a highly customizable environment involved in advanced manufacturing systems (AMS)

In order to achieve this, the study aimed at designing and developing a low-cost mechatronic sensory device for use in HRC environments. The developed sensory device had to be tested and validated to ascertain its satisfactory performance and its cost had to be justified. For this purpose, a sensory glove was developed capable of capturing the motion of a human hand.

This is a critical safety issue in HRC environments. The sensory glove was tested using a collaborative robot known as cobot. The main performance parameters were identified as the repeatability and accuracy of the sensory glove. Section 2 focuses on the literature of human–robot collaborative systems and the state of research on sensory gloves. The design and development of the sensory glove is explained in Section 3. Section 4 examines the performance of the glove with reference to other devices with a similar function. The final Section summaries the research results and suggests improvements for further work.

## 2. Literature Review

In order to design and develop an innovative sensory glove, an extensive study is needed to assess the state of research on this subject and, in particular, the issues involved in human–robot interactions as discussed in [20].

This includes information on current sensory gloves used primarily to track the hand motion and orientation of the human hand to identify the key components that make up such a device and to determine the specifications required for a low-cost sensory glove.

### 2.1. Sensory Glove

Sensory gloves are classified as a motion tracking system. They involve the integration of IMU sensors, human kinematic modelling, a microprocessor, computer hardware and software. The components are used to control the flow and processing of the data from the sensory glove. The data are passed through an algorithm to produce relevant data on the motion of the human hand for the end-user. Sensory gloves include electrical and electronic components, a computer component and a control component, which controls the data flow from the sensory glove to the computer system. These components define the foundation of a mechatronics system, shown in Figure 1, which is further expanded to accommodate the complex integration required between the components of sensory gloves. The integration of the four components require a synergistic relationship [17].

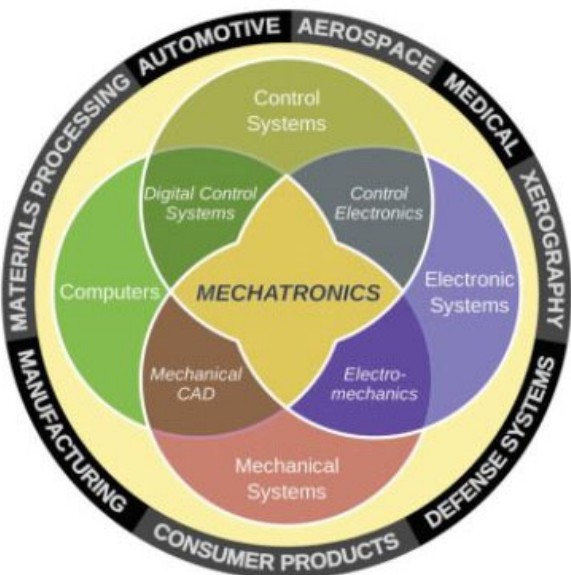

**Figure 1.** The fundamental components of a mechatronic system [21].

Extensive research has been conducted on a wide range of sensory gloves that have been designed, tested and built for academic and commercial uses. These included sensory gloves used in rehabilitation research and motion capturing for animation. Sensory gloves have grown in industries that focus on the rehabilitation of hand movement, gesture to text recognition for the deaf and hand motion tracking for computer games. A review of the subject is given in [22].

The primary goal of sensory gloves is to track the orientation and motion of a human hand. Therefore, sensory gloves have seen increased development due to the potential performance that can be achieved. A review of the research in this field is examined below. This is to gain a greater understanding of sensory gloves and how one can develop a sensory glove for an advanced manufacturing environment. The author believes that sensory gloves could act as enablers for HRC systems due to the recent advancements made and the potential benefits that it has over camera-based solutions.

Researchers in [23] designed and developed a wearable sensory glove for enhanced sensing and touching purposes for the gaming or rehabilitation industry. It was based on IMU sensors for hand tracking and introduced components for cutaneous force feedback. The glove was designed with eleven magnetic, angular rate and gravity (MARG) sensors. These sensors were positioned on the middle phalanges and proximal phalanges on each finger. One sensor was positioned on the palm of the hand to establish a reference frame. The glove was able to achieve a 95% confidence interval and an orientation estimation error of $3.06° \pm 0.12°$. It was noted that further work could be conducted by adding an accelerometer and gyroscopic sensors on the distal phalanges to improve the performance of the device. The use of MARG sensors would not be applicable in this research as the

magnetometers would interfere with the magnet fields in industrial machinery. Furthermore, the researcher noted that the use of IMU sensors on the distal phalange could lead to greater results. The sensory glove can be seen in Figure 2.

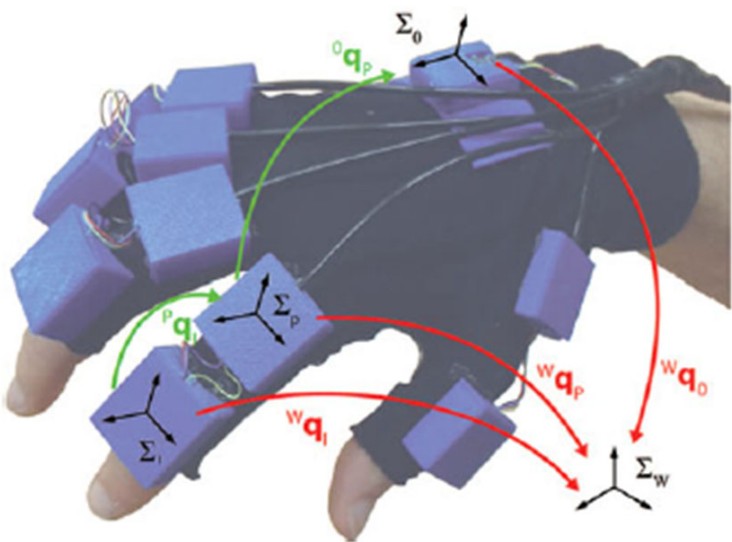

**Figure 2.** Sensory glove developed in [23].

Researchers in [24] looked to develop an on-body assessment system that allows the measurement of movements and interaction forces of the hand, fingers and thumb. It used IMU sensors to track the motion of a user's hand. The glove was designed with sixteen IMU sensors and an Atmel XMEGA microprocessor. The microprocessor was used for data acquisition and data processing algorithms. A comprehensive hand kinematic model was designed for the computer system and a robust extended Kalman filter was developed for the control system of the sensory glove. These components were used to minimize the orientation estimation error of the system. This resulted in a high dynamic range (116-degree full range finger movements per second) and a repeatability range of approximately 2 degrees.

The device displayed high performance while achieving a compact and efficient design. The components that were used to create the glove were cost-effective but the processing power needed for such a device was computationally expensive. This computational power was achieved by the glove tethered to a computer system. This was not applicable for this research as the projected aim was to create a wireless system to be used in a manufacturing environment. The lack of protection around the device was a concern as it would not be viable in a manufacturing environment, which was the application of this research. Therefore, due to the reason mentioned, it was noted that the design of the sensory glove was not applicable for this research. It was noted that a less computationally expensive solution would need to be investigated to further develop the real-time application of such a device. The sensory glove can be seen in Figure 3.

These sensory gloves display similar characteristics to those required for tracking the motion of a human hand in a human–robot collaborative environment. However, the applications of the sensory gloves were not applicable to this research. The sensory glove developed by [23] provided a protective casing for each sensor. This was noted by the researcher as a viable solution in the manufacturing application. It would ensure that the sensors would be protected in a dynamic environment such as a manufacturing floor. The performance parameters achieved by [24] were noted by the researcher as they showed the potential of such a sensory glove. It presented a potentially safe wearable solution for a worker in an HRC environment. While the system architecture was ideal for the sensory glove, the wired approach as well as the unprotected nature of the glove made it nonviable.

Therefore, the fundamental principles employed in [23,24] were combined and used to develop the sensory glove with design improvements to ensure it could be used in an HRC system.

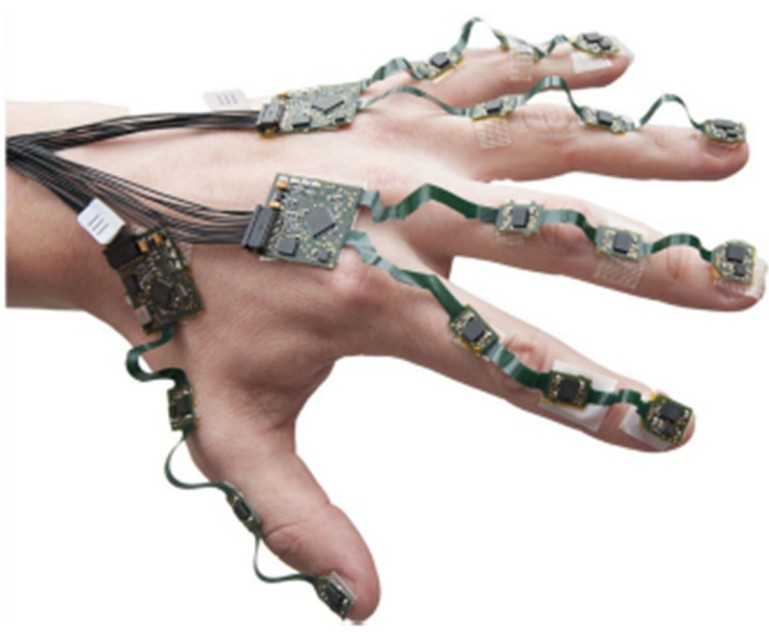

**Figure 3.** Design of sensory glove [24].

### 2.2. Human–Robot Collaboration

Human–robot collaboration is a broad ideology that is defined on three main principles according to [13]. It is suggested by researchers, such as [25], that HRC systems include not only the sharing of a physical workspace but also a common task. HRC systems can be achieved if the absolute safety of the worker can be guaranteed and accomplished. Accordingly, a framework can be developed that consists of three levels of interaction between the robot and the human worker as shown in Figure 4.

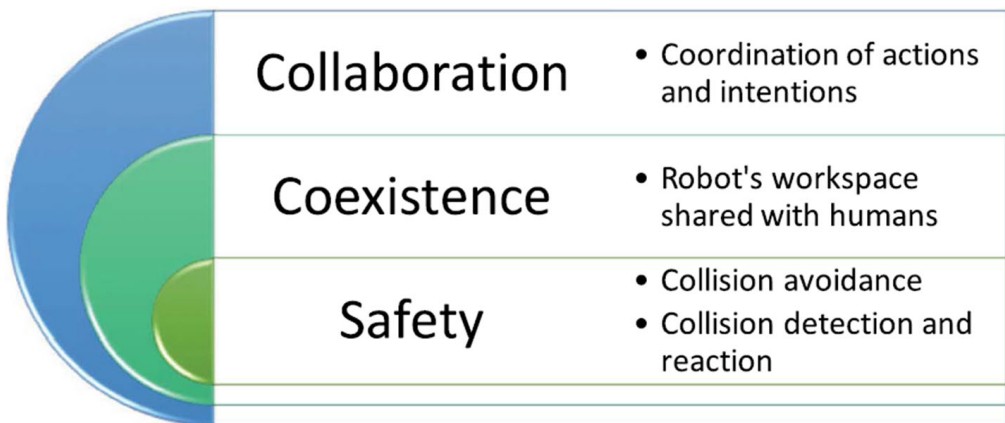

**Figure 4.** Levels of interaction framework for HRC systems [5].

The framework indicates that any level of interaction must include the features associated with it as well as the features of the previous level interaction. The framework shows that safety is essential when collaboration exists in an HRC system, as a critical feature to achieve. This can be performed by integrating systems such as sensory equipment, external mechanics and path avoidance algorithms. Sensory gloves, as shown in the literature above, have the potential to provide the required level of safety to enable HRC systems to operate reliably by meeting specific performance parameters.

An example of such an HRC system is a recent EU project known as ROBO-PARTNER, which aimed at creating a seamless HRC system for safe operations in advanced assembly factories of the future. The project used the foundation of cobots and the principles laid out by [26]. The project's initial research suggested that human skill was the main driver towards high added-value products and the integration of a robot's strength, precision and repeatability would improve it [27]. In order to create a safe and productive workspace, an integration with autonomous robots through a user-friendly interface was required. The project outlined that a human–robot cooperative assembly operation could be possible through enhanced sensor-based interaction. The sensor-based interaction could be achieved with the use of a device such as a sensory glove. It would provide the hardware and software for data acquisition, data processing and motion tracking of the hand motion in real time. The description of the ROBO-PARTNER project can be seen in Figure 5.

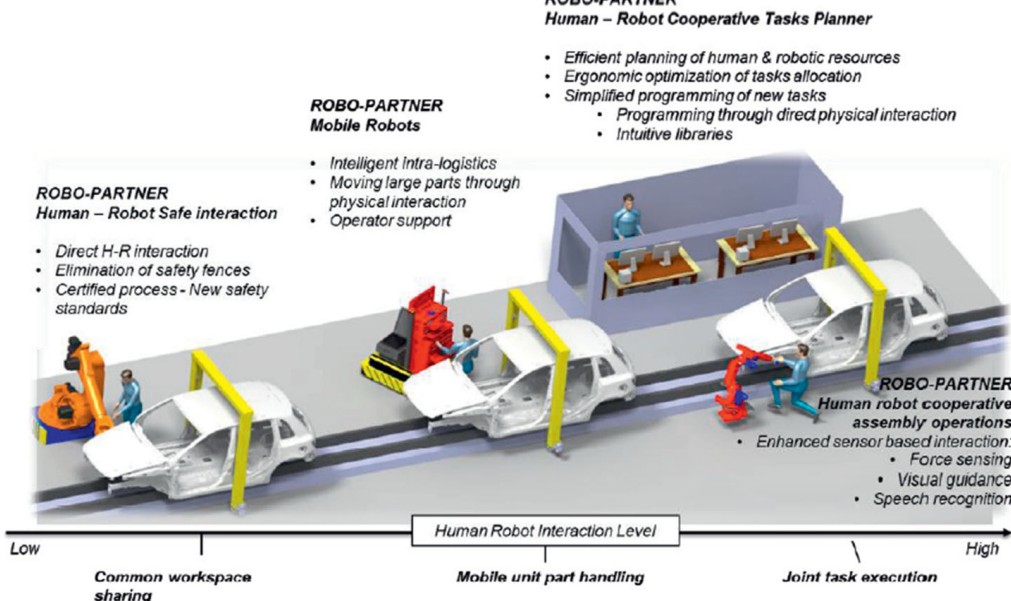

**Figure 5.** ROBO-PARTNER production paradigm in the automotive industry [27].

Based on the ROBO-PARTNER, the sensory glove could allow the human to feed certain commands, based on the hand's orientation, as to what action was to be performed by the robot. A certain hand orientation could command the robot to stop so a tool change could be performed. The robot would then be able to stop in its position for the human to interact with it. These methods are explored in the results section.

These systems are what are required in the advanced manufacturing sectors as a paradigm shift has occurred in the production industry. This shift has been from the mass production of generic products to the mass production of customizable products [26]. This type of mass production requires industries to be highly reconfigurable and flexible. There have been significant research advancements in FMS. These systems have been designed to allow for easy adaption to changes in the type and quantity of a product being manufactured [19]. The main drawback of such systems is the substantial setup cost.

In order to achieve mass customization, reference [26] suggests that it will be necessary to combine high reconfigurability with larger degrees of automation. This goal can be achieved through the development and establishment of HRC systems as they act as an enabler. Therefore, sensory gloves could be used as a tool to enable HRC systems. With the creation of this HRC system, the production process leverages the dexterity, flexibility and decision-making capability of a human to the speed, precision and power of the robot. According to [28], industry requires this to achieve mass customization. HRC systems research shows that most robots that have HRC capabilities come at a substantial cost. Other sectors, such as the automotive industry, also benefit from using sensory gloves

as they significantly improve productivity and lower the setup cost of FMS within the advanced manufacturing sector.

## 3. Mechatronic Design and Assembly of Sensory Glove

Through a design process, different approaches were developed to create a sensory glove system. However, the manufacturing and assembly processes led to a number of engineering challenges. These problems are the result of multiple factors including hardware limitations, integration configurations and random errors. These all contributed to the process of manufacturing as well as validating the effectiveness of the sensory glove. The manufacturing and assembly processes are divided into four subsections. These subsections include:

- Electrical and Electronic System.
- Control System.
- Computer System.
- Mechanical System.

The subsections of the system are used to create and identify the most cost-effective and efficient solution. Cost-effectiveness revolved around the use of relatively cheap components with efficient design and power parameters. Each element of the system was manufactured and integrated into the mechatronic system. The components needed to work efficiently as an integrated system to ensure a low-power efficiency and performance. Challenges in the process of producing the sensory glove are outlined in the present chapter. Once the systems were created, the final product was assembled and tested.

### 3.1. Design and Assembly Process of Sensory Glove

3.1.1. The Electrical and Electronic System

The electrical and electronic architecture of the sensory glove included fifteen six-axis accelerometer and gyroscopic sensors (MPU6050), two multiplexers and an Arduino© Due from DIY Electronics in Durban, South Africa. The fifteen MPU6050s were not able to be connected to a single Arduino© Due without a multiplexer device. Therefore, two multiplexers were used as it was a three-line to an eight-line decoder. This enabled eight MPU6050s to be connected to one multiplexer, while the remaining seven connected to the second multiplexer. The multiplexers had three controller pins, which allowed the Arduino© to control the data from each sensor as it cycled through them. The multiplexers had three enabler pins. This allowed the Arduino© Due to switch between the sensors. The multiplexers and MPU6050 sensors were all powered by the same 3.3 V supplier. This was a requirement as the device needed to achieve a power-efficient structure.

Figure 6 represents a simplified circuitry of the sensory glove system. It shows one MPU6050 and a multiplexer connected to an Arduino© Due. The figure was used to illustrate how the components were connected together through the multiplexers and the Arduino©.

In order to create an energy-efficient and compact circuitry to connect all the electronic components, a Printed Circuit Board (PCB) was created. The PCB created a more compact electrical and electronic architecture for the system. The PCB can be seen in Figure 7.

The PCB was designed for the footprint of an Arduino© Due to ensure an easy connection and installation process. The board's configuration copied the designed circuit, which had been created on the breadboard for the system. Three LEDs were installed to visually validate the programming code. The code logic enabled the selection and deselection of the output lines on the multiplexers.

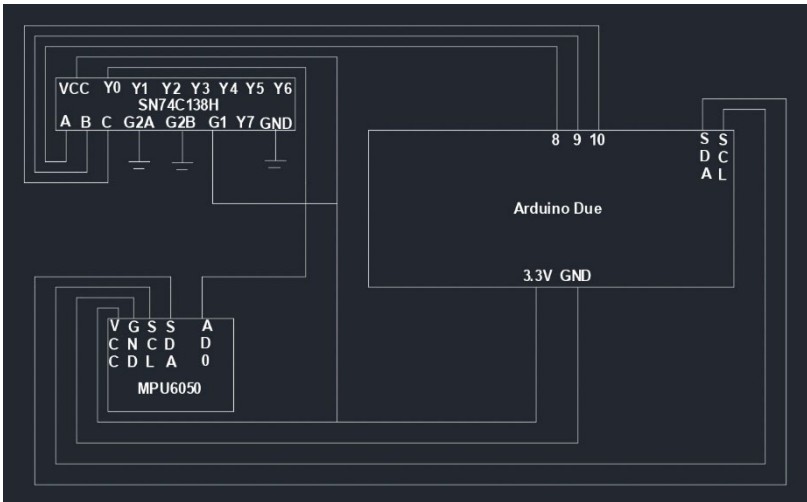

**Figure 6.** Sketch of electrical and electronic architecture.

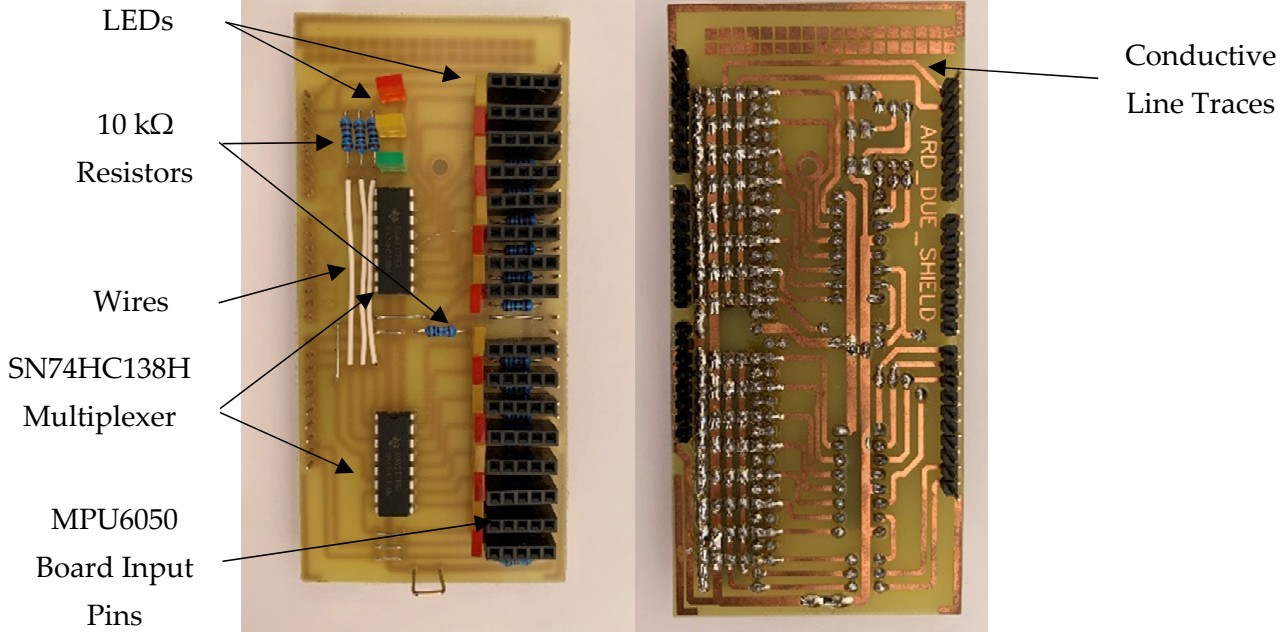

**Figure 7.** Design of modified PCB: top view (**Left**) and bottom view (**Right**).

3.1.2. The Computer System

In order to validate and test the performance of the glove, a three-dimensional model of the hand was created to use in a simulated testing environment. The three-dimensional model was created using SolidWorks, a solid modelling computer-aided design software. The hand model was designed in two parts. The first part consisted of the palm, while the second part included the fingers. The dimensions of the model were based on studies that were conducted to measure the average human hand dimensions as discussed in [29].

The palm was created first and it was designed as a semi-spherical shape with a rectangular footprint. It was modelled as the palm, as it best captured the centre of gravity and moment of inertia. The palm was designed with five-link connectors, where the fingers could be attached. The link connectors resembled a one-dimensional joint between the palm and each finger.

Once the palm was completed, the design of the finger began. The structure of the finger is made of three prominent bones, which are known as the distal phalanges, middle phalanges and proximal phalanges. These bones begin at the knuckle of the hand and

extend to the fingertip. Therefore, each phalange in the finger was designed as rectangular blocks with link connectors on either side. The link connectors were used to model the DOF joints between adjacent phalanges. The phalanges were developed in this configuration to represent the first and second moments of inertia accurately.

When all the parts of the hand model were developed, the parts were integrated into a hand assembly. All the phalanges were designed with joint connectors to allow for easy assembly and to represent the one DOF joint between each phalange in the finger structure. The final assembly of the human hand model is shown in Figure 8.

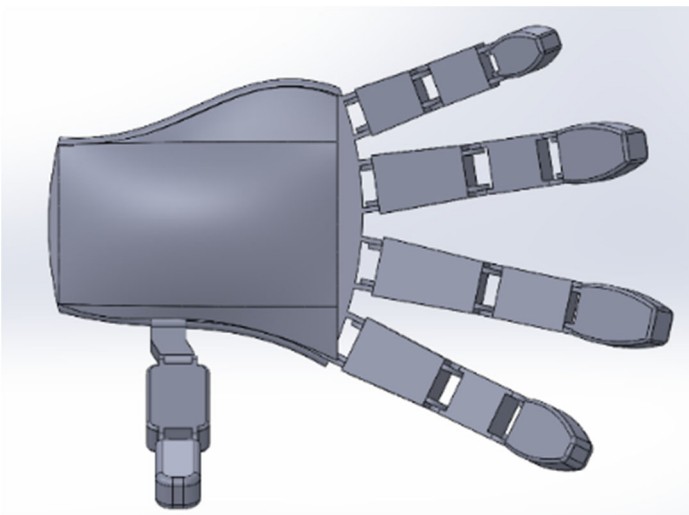

**Figure 8.** Mechanical assembly of a human hand.

With the hand model complete, the model could be imported into a simulated testing environment for performance analysis. This environment was created using Simulink©, which is a graphical programming environment for simulating and the analysis of dynamic systems. Simulink© had the capability of integrating the CAD model with the control system sensory glove. This was achieved with the use of the Simscape multibody tool. Simulink© also had advanced Arduino© integration capabilities. Once imported, the tool formulated and solved the equations of motion and the dynamics of the system. The package provided a platform to effectively integrate the mechanical model of the hand with the control system of the sensory glove.

The Simscape multibody tool modelled the phalanges as links. The joints between adjacent phalanges were modelled as one DOF revolute joint. Therefore, the package created a single kinematic chain system controlled through torque or motion inputs of each revolute joint. This was ideal as research has shown that the best way to model the human hand is as a straight chain link between the wrist and fingertips [30]. The model was designed to be controlled through motion inputs to simulate the motion of the human hand model. Figure 9 shows the model of the human hand created using the Simscape multibody tool in Simulink©.

### 3.1.3. The Control System

In order to design the control system for the project, Simulink© was used. The control system design started with the data acquisition process. The data acquisition process involved performing multiple operations on sensor data from MPU6050 sensors with the use of a control algorithm. The algorithm would capture the raw data from the sensors and configure it to an acceleration and gyroscopic value for the sensors. Once the data were configured, roll and pitch values for each sensor were calculated and recorded. A simplified version of the process can be seen in Figure 10 as it illustrates the data acquisition and configuration process of the sensor's data.

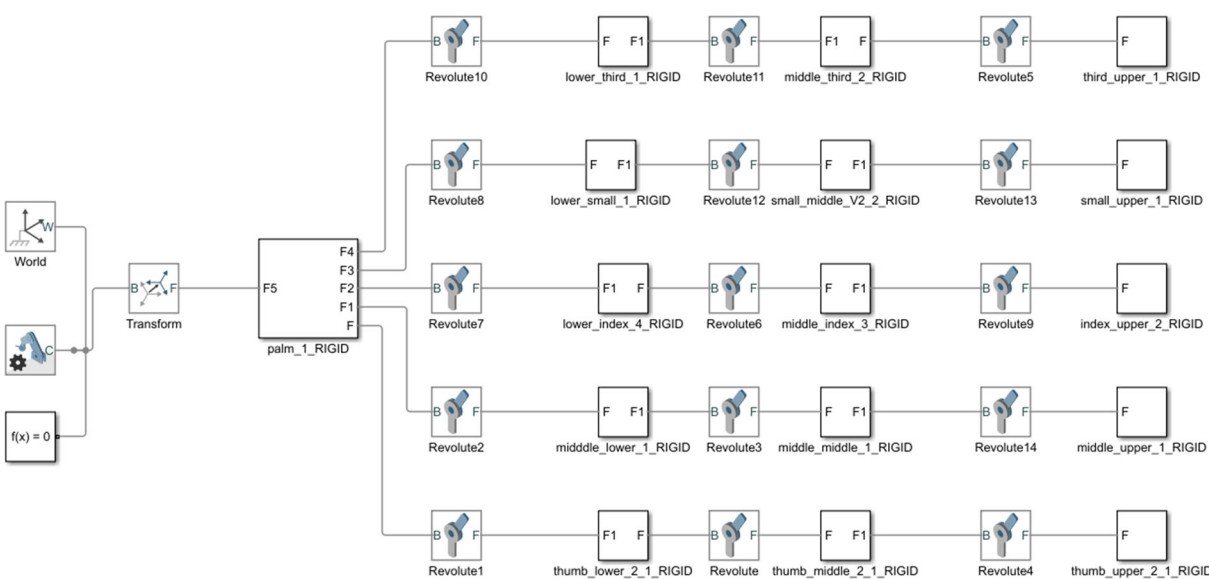

**Figure 9.** Simscape multibody model of the human hand.

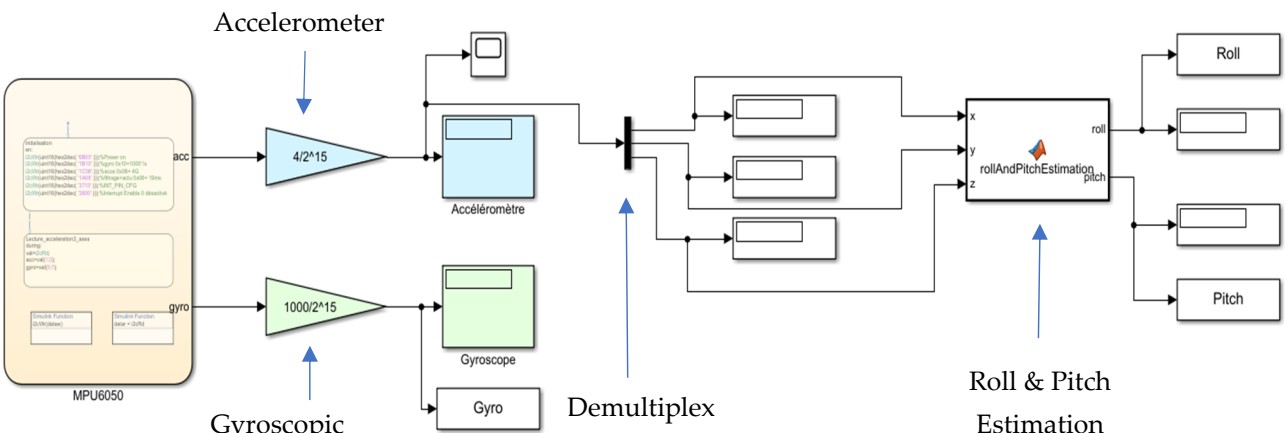

**Figure 10.** Simplified data acquisition and configuration model.

Initially, the calculated roll and pitch values of the sensor were to be integrated to provide further accuracy for the orientation of the sensors. However, during the initial stage, it was discovered that while this process could work when dealing with a single sensor, it would not work in the case of multiple sensors due to the time delay in processing the data (approximately 80 milliseconds). Therefore, the pitch values were then exported into the next stage of the control system.

Once the data acquisition process was completed, the values were imported into a post-processing model. The post-processing model configured the data for the human hand model in Simulink© and began designing an algorithm that 'cleaned' the data by eliminating process noise in the signal (known as a cleaning function). Once the data were 'cleaned', an algorithm was written to constrain the dynamic range of each finger joint (known as the data prep function). Lastly, the data were fed through an algorithm that ensured that each sensor was configured to the correct phalange on the finger structure (known as the finger alignment function). This was carried out by creating an equation to calculate the relative angle of a joint between two phalanges. This calculation used the sensor on the palm of the hand as a reference angle for all finger structures. Each bone on the finger structure, as you move from the proximal phalange towards the distal phalange, used the previous angular value and its value to determine its orientation.

Once these configurations had been completed, the data were ready to be integrated into the computer system of the sensory glove. In order to achieve this, the data from all

the MPU6050 sensors (three sensors per finger) were integrated into the Simulink© hand model. The integration into the hand model was complex as it involved second-order Simulink© functions as well as a filtering process. The filtering process was essential as it further reduced the process noise in the sensor's data.

The filtering process involved an averaging filter. An averaging filter was chosen as it required less computational power and it was ideal for real-time response systems. Figure 11 shows the data from the sensors before the averaging filter was implemented (blue line) and after the averaging filter was implemented (yellow line).

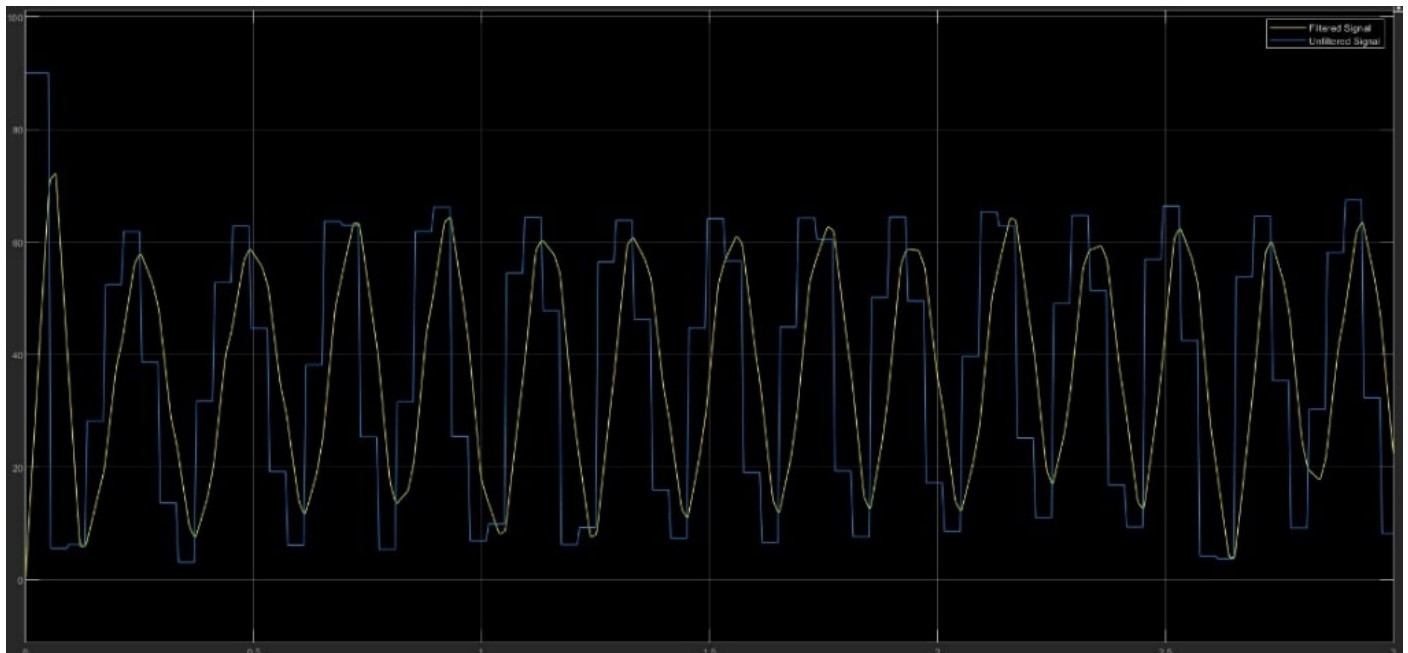

**Figure 11.** Sensor data before (blue line) and after (yellow line) the averaging filter was implemented.

With the filtering implemented, the data were passed through two gain blocks. The first gain block converted the angular displacement of the rotation from degree to radians. The second gain block was the correction orientation block. The data were finally fed through a PS-Simulink converter in order to integrate the data with the Simulink© human hand model. This system model can be seen in Figure 12, which represents part 1 of the post-processing model in Simulink©.

All the data from the sensors were integrated with the respective joints to control the phalanges on the human hand model. This enabled the model to simulate the motion of the sensory glove as it moved. The phalanges were controlled by the input sensor data of the joints. The pitch data were fed into the respective joints as they had one degree of freedom. This can be seen in Figure 13, which shows part 2 of the post-processing model in Simulink©. The final design of the Simulink© hand model is shown in Figure 14. It displays the entire Simulink© hand model and how the post-processing model in Simulink© and the hand model were integrated together.

### 3.1.4. The Mechanical System

The mechanical system for the sensory glove included the design and manufacture of the glove architecture. The glove was made of a low-cost material as the goal of the project was to produce a low-cost glove. Based on the hardware and cost requirements for the device, a Mac Afric working glove was purchased. It would provide the base for the sensory glove to be built on as it was made from fleeced cotton with latex coating. This ensured that it was lightweight and malleable.

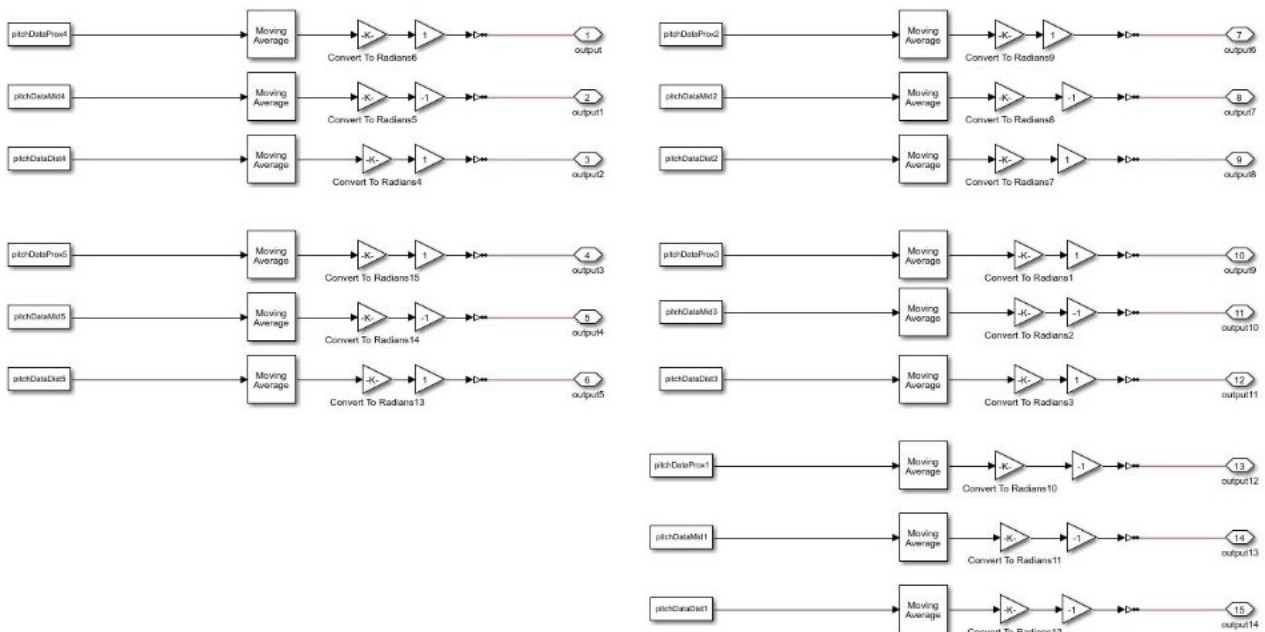

**Figure 12.** Part 1 of the post-processing model in Simulink©.

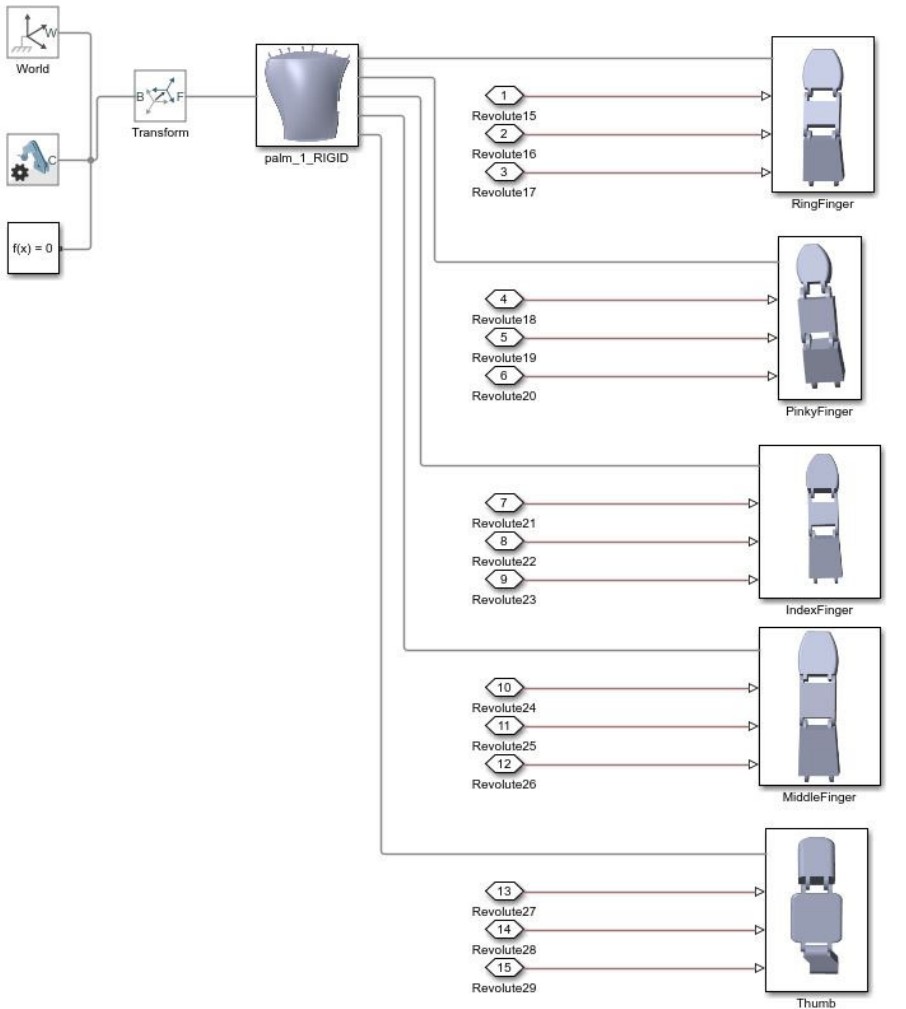

**Figure 13.** Post-processing model (Part 2).

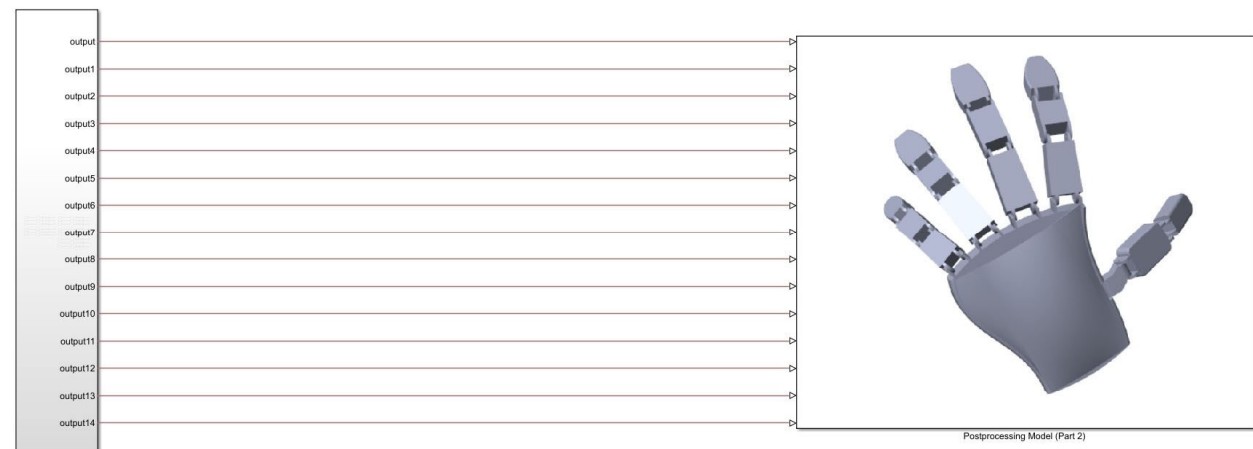

**Figure 14.** Entire Simulink© model.

To attach the sensors to the Mac Afric glove, sensor holders needed to be designed and manufactured. These sensor holders were customized to the dimensions of the sensors and the dimensions of the fingers. These holders would provide support for the sensors to operate while protecting them in operation. The sensor holders also required a low-cost manufacturing method to ensure that costs were minimal. Therefore, it was decided that the sensor holders would be 3D printed to ensure a custom, light and low-cost solution. The sensors holders were designed to hold the sensors and fit around a person's finger to ensure a secure and stable position. The base of the sensor holders was designed to the curvature of the phalanges. The top of the holders was designed to hold and protect the sensors. The SolidWorks© design of the sensor holders can be seen in Figure 15.

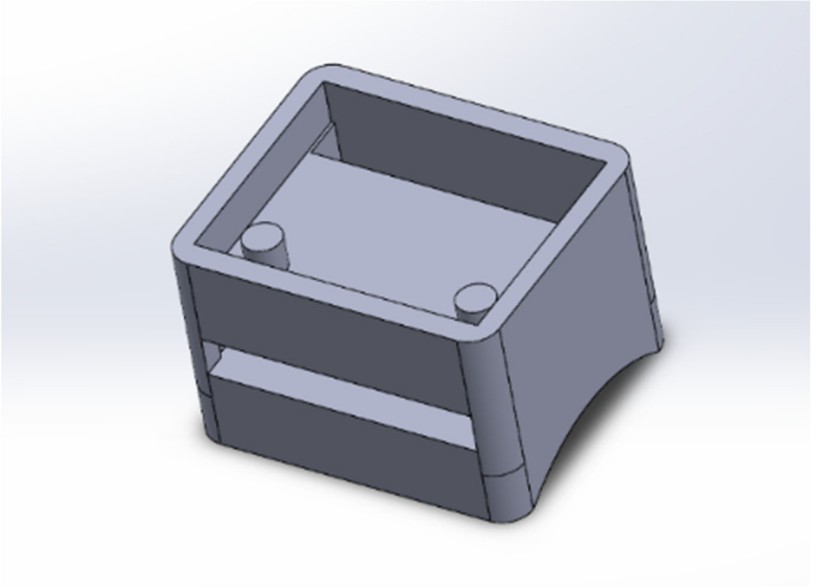

**Figure 15.** Final design of sensor holder.

To attach the sensors to the glove, alterations were required. The alterations included sewing Velcro patches to the glove. The hook side of the Velcro was sewn onto the glove. The velvet side was attached to the sensor holders.

3.1.5. Conclusions

Once all the subsystems for the sensory glove were designed and developed, the sensory glove was assembled. With all the components assembled and systems integrated,

the sensory glove encompassed the four key elements of a mechatronic system. Each subsystem was designed to ensure that it could integrate together. This design improved on the sensory gloves researched in this literature review as it removed magnetometer sensors to reduce interference with industrial machines and placed IMU sensors on the distal phalange to improve performance. The total weight of the sensory glove was 400 grams. This was all essential to creating an efficient, lightweight and compact solution. The total cost of the development of the sensory glove was 120 euros. This was far less than the commercially available sensory glove system developed by Sensoryx. This showed that the sensory glove was a low-cost solution. The sensory glove is shown in Figure 16.

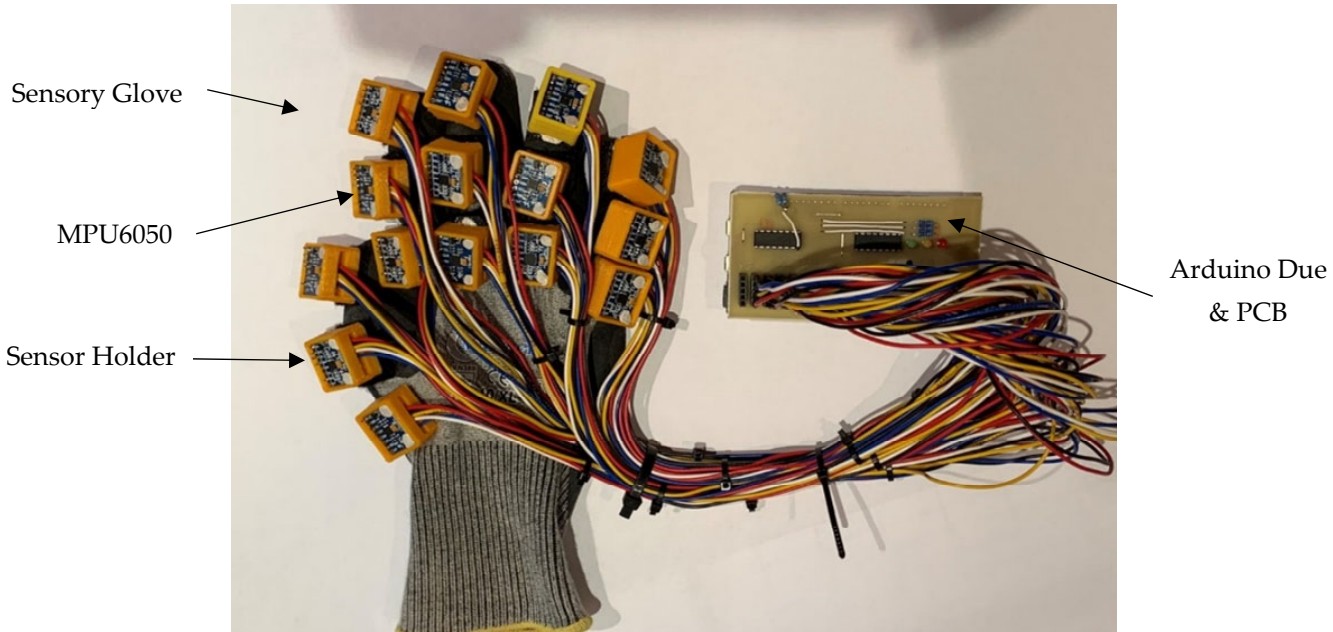

**Figure 16.** The final form of the sensory glove.

## 4. Testing and Results of Sensory Glove

To determine the performance of the sensory glove, performance metrics needed to be tested. Three performance tests were developed and performed on the device. The performance tests covered the performance metrics of the glove to ensure that the final design met the objective of the research. The present section has three sections explaining the methodology and results of each test. These tests were the accuracy test, dynamic range test and application test.

### 4.1. Accuracy Test

The aim was to determine the accuracy of the angular measurement obtained from the sensory glove while the hand was in a desired orientation. This methodology of testing the glove started by creating a 3D printed model of a human hand. The 3D printed human hand was similar to the hand model that was designed for the control system.

Once the hand model was printed, the researcher set five predetermined hand orientations to carry out the test. Each hand orientation was unique as they required specific relative angular values of each phalange to validate the accuracy of the sensory glove. This was conducted over a range of values. The researcher developed a procedure to perform the test and tabulate the results. The results of the test can be seen in Table 1.

The researcher used a Goniometer, which is a handheld device used to measure angles between two adjacent objects. It was used to measure the actual values between adjacent phalanges on a finger. The table showed the measured relative angular values using the Goniometer, as well as the simulated values from the sensory glove for the index finger.

There was a high level of accuracy due to an average 10 percent difference between the simulated and measured values. A Pearson's correlation coefficient indicated that there was a strong linear relationship between the two sets of data (R > 0.92) that allowed the average difference to be calculated. The Pearson's correlation coefficient can be seen in Table 2.

**Table 1.** Measured relative angular value versus simulated values from the sensory glove for the index finger.

| Data Type | Trial | 1 | 2 | 3 | 4 | 5 |
|---|---|---|---|---|---|---|
| Goniometer Values (°) | Relative Proximal Joint Value | 1.0 | 4.0 | 9.0 | 28.0 | 60 |
| | Relative Middle Joint Value | 4.0 | 13.0 | 19.0 | 40.0 | 42 |
| | Relative Distal Joint Value | 1.0 | 1.0 | 3.0 | 8.0 | 32 |
| Sensory Glove Values (°) | Relative Proximal Joint Value | 1.1 | 4.3 | 9.3 | 28.4 | 60.5 |
| | Relative Middle Joint Value | 3.9 | 13.2 | 18.8 | 40.4 | 42.5 |
| | Relative Distal Joint Value | 1.2 | 1.3 | 3.3 | 8.4 | 32.5 |

**Table 2.** Pearson's coefficient for index finger joint values.

| Digit | Relative Proximal Joint | Relative Middle Joint | Relative Distal Joint |
|---|---|---|---|
| Pearson's Coefficient | 0.9205 | 0.9259 | 0.9423 |

While the Person's correlation shows that there was a strong linear relationship between the two datasets, it does not provide enough evidence that the two datasets are statistically comparable. Therefore, the 'limit of agreement' technique was used. This technique is based on the Bland–Altman method and has been cited on over 11,500 occasions [31]. The Bland–Altman method calculates the mean difference between the two measured datasets and a 95% limit of agreement as the mean difference (2 standard deviations from the mean) [31]. If 95% of the differences lie within the 95% limits, then the two datasets are said to be statistically comparable.

Therefore, the researcher used this method to determine if the two datasets were statistically comparable. Figure 17 shows a Bland–Altman agreement graph that used the theory from the 'limit of agreement' on the relative angular values from the index finger. The graph showed that 95% of the differences lay within the 95% limits. Therefore, the results of the accuracy test showed that the sensory glove was an effective and accurate solution to determine the orientation of the human hand.

### 4.2. Dynamic Range Test

The aim was to determine the dynamic range performance metric for the sensory glove system. This allowed the researcher to establish the speed and reliability of the system. The reliability of the sensory glove was essential to ensure that the system could provide consistent and accurate readings over a period of time in an HRC environment. Therefore, the test provided data on how the sensory glove could improve the manufacturing environment's safety. With safety as a constraint in the development of the HRC systems, it had the potential to act as a catalyst in the field of research.

This test was performed by repeating thirty flexion–extension movements with all joints on the hand. This did not track the abduction of the joints due to the design requirements of the system. The flexion–extension action had to be consistent each time the test was performed. In order to achieve this, modelling clay was used to create a cast to ensure that the same hand motion occurred each time the test was performed. The use of a metronome device was used to ensure the speed of the motion was kept constant. A metronome is a mechanical instrument that makes repeated clicking or beeps at an adjustable speed [32]. The metronome tempo was set at approximately 110 BPM. The cast and the metronome can be seen in Figure 18.

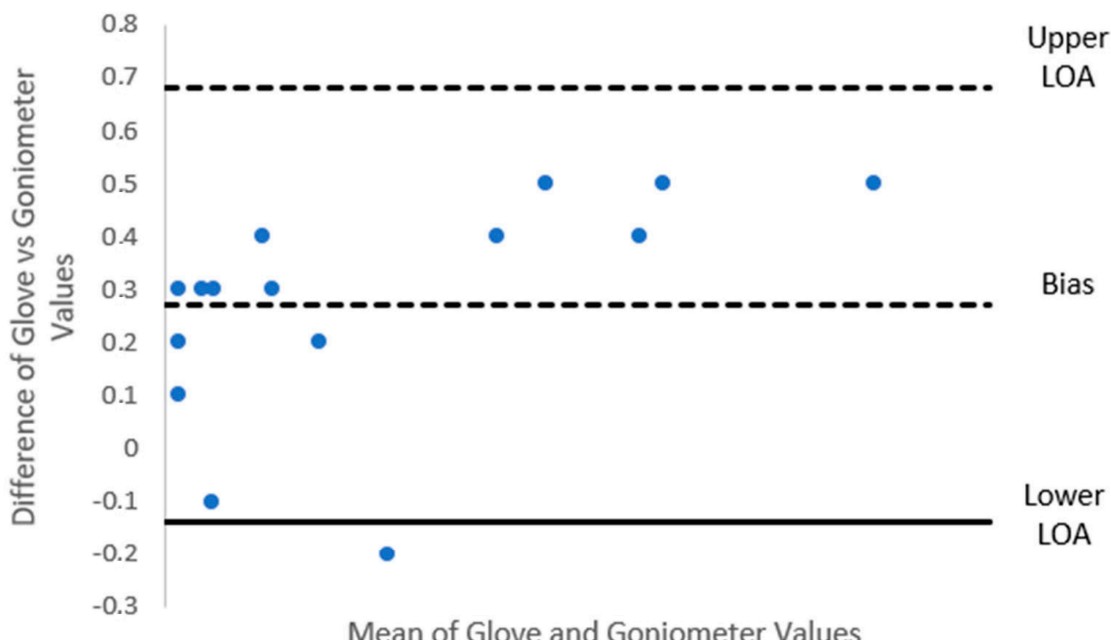

**Figure 17.** A Bland−Altman agreement graph.

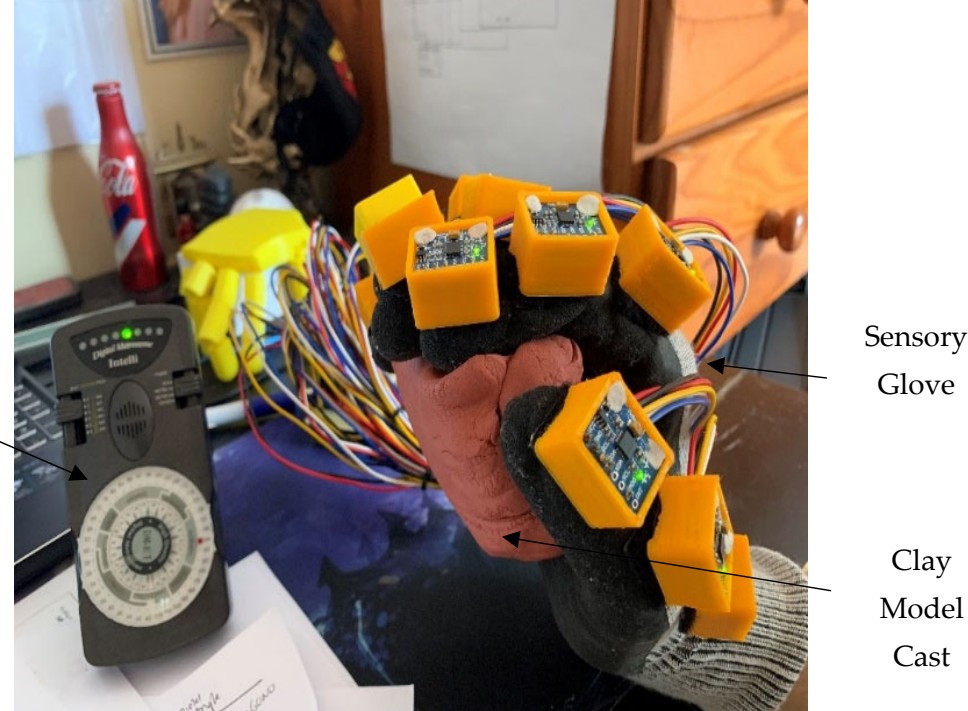

**Figure 18.** Setup of Dynamic Range Test with a metronome, sensory glove and clay model cast.

During the test, the relative angles of the joints between the adjacent phalanges were measured. All fingers performed a motion that experienced little to no abduction motion. In the beginning of the test, the researcher's hand was required to be flat and perpendicular to the surface of the earth. A procedure was designed to carry out the test. This procedure included the user operating the glove between two predetermined orientations at the desired speed. This allowed the readings from the sensory glove to be recorded and tabulated to validate and verify its performance. For the purpose of the test, the motion of

the index finger was analysed. This was conducted to perform an extensive analysis of the responsive nature of the system. The results are shown in Figure 19.

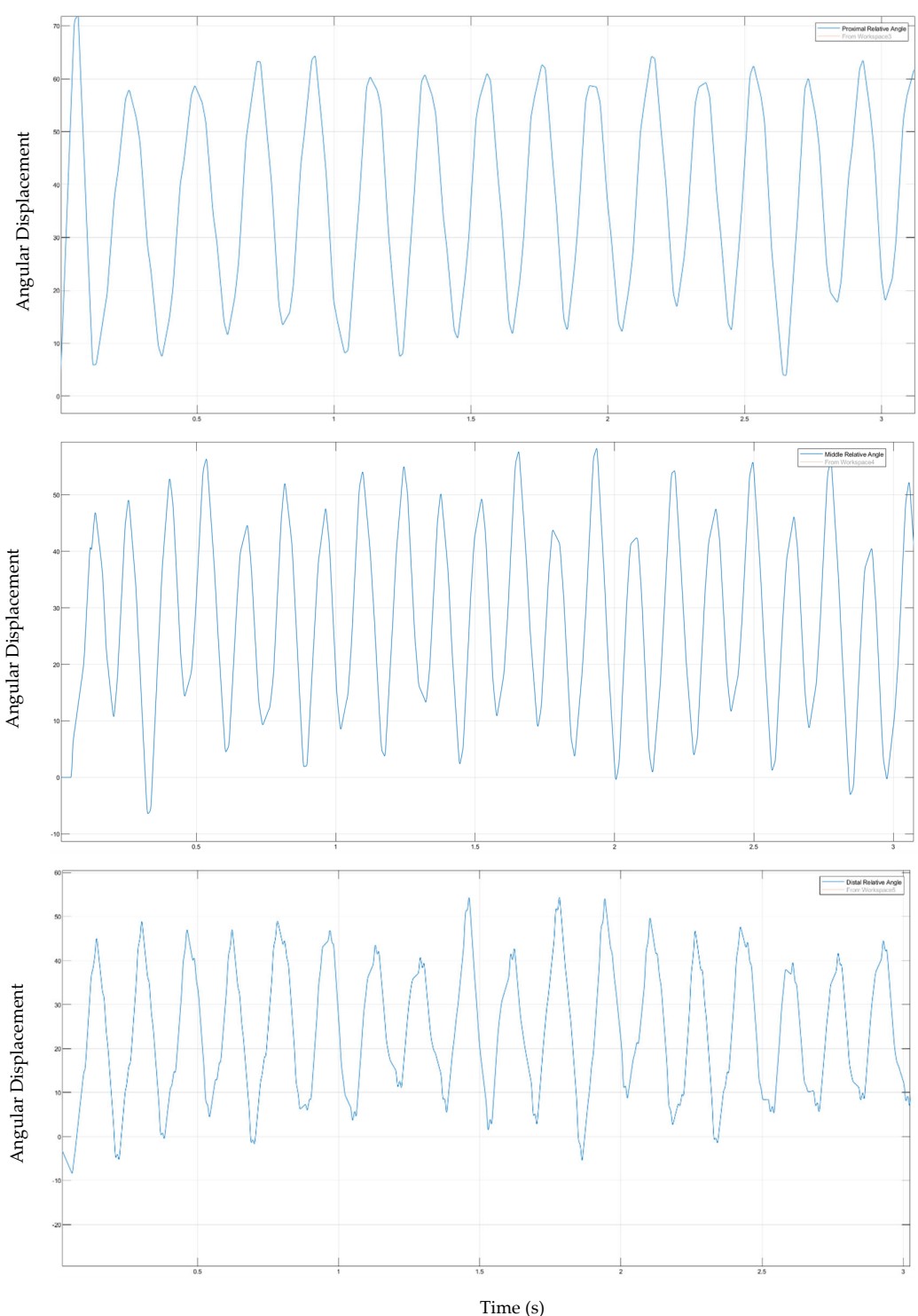

**Figure 19.** Dynamic range of the proximal joint (**Top**), middle joint (**Middle**) and distal joint (**Bottom**) on the sensory glove.

It was noted that in order to approximate the dynamic range of the finger joints, the beginning of each joint cycle occurred when the derivative of the angular velocity was zero. Therefore, the dynamic range was calculated between the time interval of two consecutive points, where the derivative of the angular velocity was zero. The bandwidth of the filter

was large enough to track the rapid movements of the system; however, the metronome tempo was not at the correct speed to capture all the motion. The tempo was chosen based on the sampling time of the filter. The problem was that since three sensors were connected to a finger and they were read individually, there was a time delay between reading the first sensor on the proximal phalange on the finger structure and the last sensor on the distal phalange. Therefore, the tempo had to be slower. Due to this, the metronome's tempo was reduced by approximately a third of the original value.

The result showed each joint's dynamic range on the index finger during the first three seconds in the experiment. The dynamic range of the proximal, middle and distal joints was approximately fifty degrees, forty degrees and forty degrees, respectively. The dynamic range of the sensors was significantly less than the static constraints on the respective finger joints of the index finger that are outlined by [29]. Overall, the results showed that the sensory glove had a satisfactory performance for use in an HRC environment.

### 4.3. Application Test

In order to determine the effectiveness of the device in an HRC environment, an application test was developed. This test was conducted with the use of an xArm 5 Lite. This type of robotic arm was used as they are standard pieces of equipment on production and manufacturing floors. The researcher designed an algorithm that would compare a predetermined hand orientation of the sensory glove to one that was stored in the computer system. The predetermined hand orientation was determined by the Simulink control system. If the orientation matched, then an action or operation would start. A more detailed explanation of the algorithm can be seen in Figure 20. The test used the sensory glove to change the start/end position of the robot arm while in operation. This was based on the part received by the robot. The use of the sensory glove showed it was possible to operate the robot in a collaborative environment. A procedure was designed to carry out the test and results were tabulated.

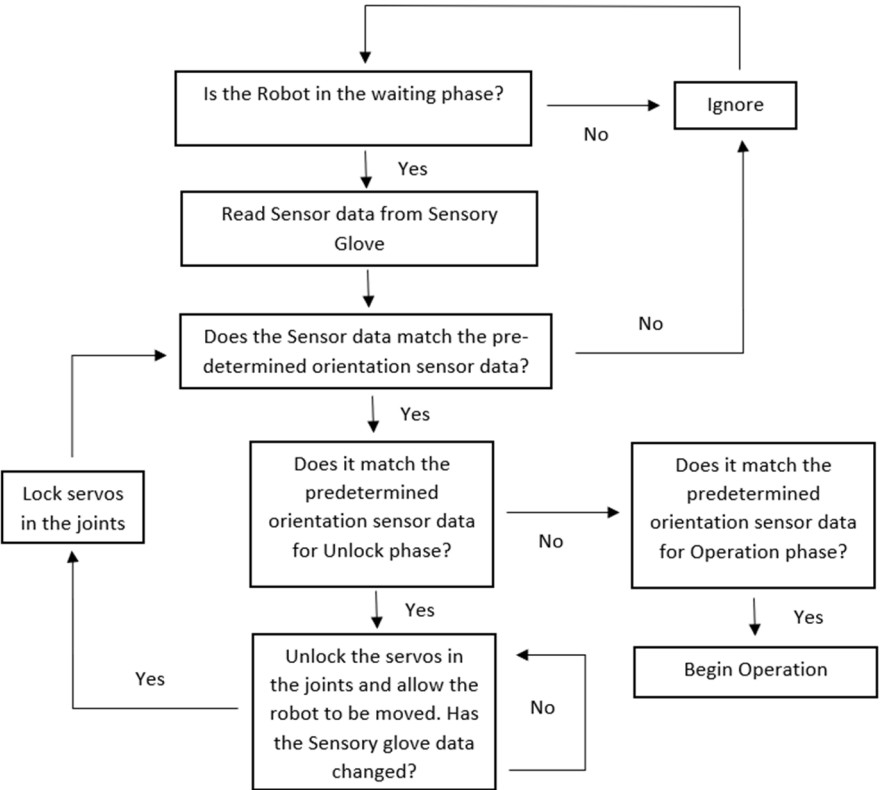

**Figure 20.** Flowchart of the designed algorithm.

Ten trials were conducted to confirm that the algorithm unlocked the robot joints when in a predetermined orientation and if the algorithm resumed the robot's operation when in a predetermined orientation. Table 3 below shows the results of the test and the efficiency of the sensory glove.

**Table 3.** Results from HRC application test.

| Trials | 1 | 2 | 3 | 4 | 5 | 6 | 7 | 8 | 9 | 10 | Efficiency (%) |
|---|---|---|---|---|---|---|---|---|---|---|---|
| Did the Joints Unlock? | Yes | Yes | Yes | Yes | Yes | Yes | Yes | Yes | Yes | Yes | 100 |
| Did Operation Resume? | Yes | Yes | Yes | Yes | Yes | Yes | Yes | Yes | Yes | Yes | 100 |

The results showed a 100% efficiency of the system and the designed algorithm. This showed that the glove could be integrated into an HRC environment. As such, the sensory glove can provide a working environment for human–robot collaboration. Figures 21 and 22 show the sensory glove and xArm 5 Lite in collaboration during the application test.

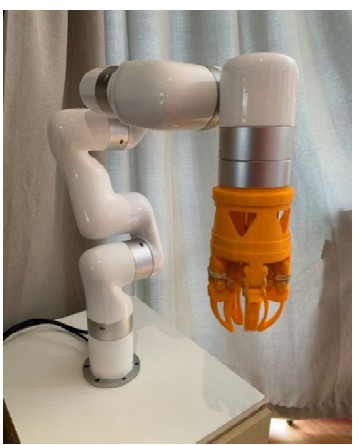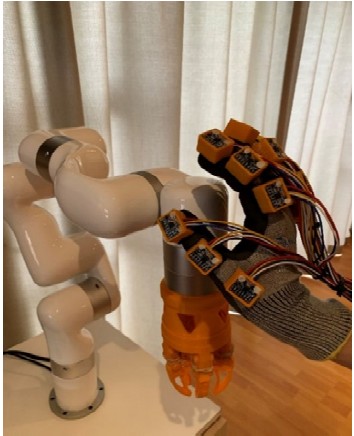

**Figure 21.** xArm 5 Lite in the waiting phase (**Left**) and xArm 5 Lite in waiting phase as sensory glove approaches the neck of the robot (**Right**).

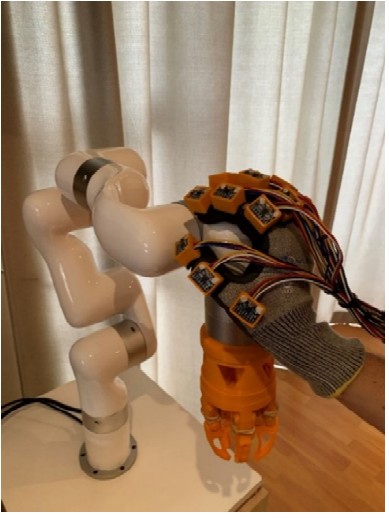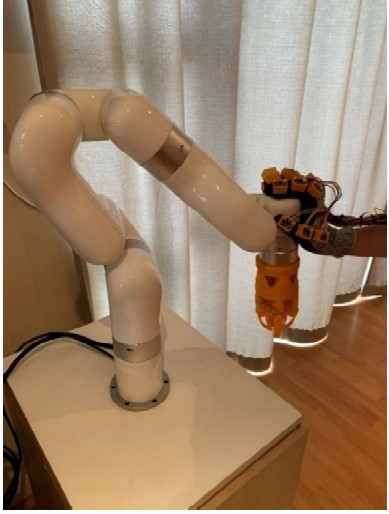

**Figure 22.** Sensory gloves unlock the joints of the xArm robotic arm to enter the unlock phase (**Left**) and the sensory glove moves the robotic arm into its final position (**Right**). Once the sensory glove releases the neck, the xArm 5 Lite will be in operation phase.

## 5. Discussion

### 5.1. Design and Development of Sensory Glove

Early in the development stage of the system, it was required that a balance between integration and performance had to be achieved. This was to ensure that the sensory glove was defined as a mechatronic device with the required performance metrics. This approach was the focus area when developing each subsystem of the sensory glove.

For the control aspect of the study, Simulink© and SolidWorks© were the ideal software packages. Simulink© provided extensive filtering design capabilities and robust libraries for mathematical equations, which enabled the creation of the control system. The Simscape multibody tool provided an effective way to integrate the SolidWorks© hand model with the Simulink© Control model. This ensured that the human hand model was integrated with the control system of the sensory glove. Furthermore, it allowed extensive testing and analysis to be conducted on the sensory glove to validate its performance and practicality.

For the electrical and electronic components of the system, the Arduino© Due provided a robust solution for the data acquisition process. Compared to similar devices, the MPU6050 sensors were an effective solution for motion capturing due to their low-cost, size, and performance capabilities. The mechanical system, which included the glove and the sensor holders, ensured that all the hardware components of the system were protected.

Through the above development of the sensory system, each element of the mechatronic system was designed and implemented to ensure an efficient, well-integrated, safe and cost-effective solution. The final product of the sensory glove is shown in Figure 23.

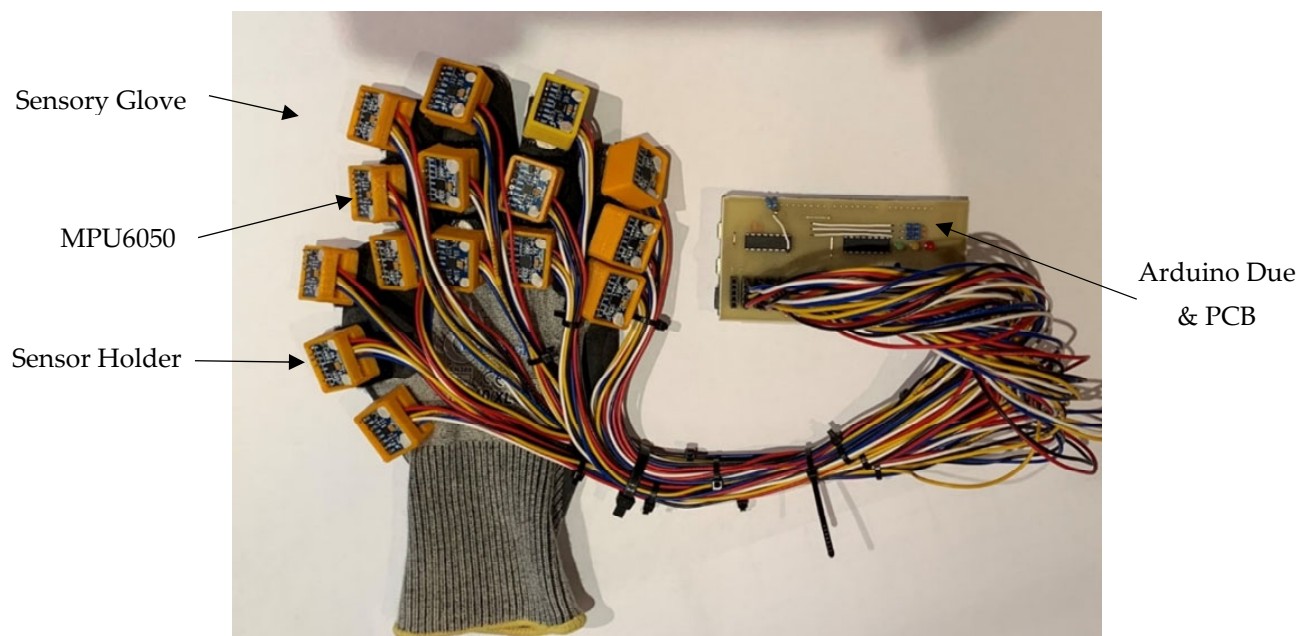

**Figure 23.** The final sensory glove.

### 5.2. Testing and Validation of Sensory Glove

In order to test the performance of the sensory glove, three tests were conducted. Each test was designed to measure the performance metrics of the glove. These tests also ensured that the sensory glove acted as an enabler for an HRC system, as outlined in the Introduction section. With the primary focus of the sensory glove being a mechatronic system, there was a need to establish a relationship between the subsystem performance and integration.

The first test aimed at determining the accuracy of the sensory glove while the hand was in a desired orientation. The results showed that the sensory glove had a high level of

accuracy due to an average of a 10 percent difference between the simulated and measured values and the Pearson's correlation coefficient of R > 0.92. This was further proven with a Bland–Altman agreement graph showing that the two methods used to determine the accuracy of the glove were comparable. This supported the correlation value and showed that the system was accurate and could effectively track the orientation of a human hand. Further work can be conducted to improve this with an advanced Extended/Unscented Kalman filter. This would only be useful if the computation costs of these filtering solutions are optimized for real-time applications.

The second test aimed at analyzing the dynamic range of the sensory glove. The dynamic range of the proximal, middle and distal joints was significantly less than the static constraints for the average human hand documented by [29]. The hand movement's tempo was set based on the sampling frequency of the system. The system was unable to capture the correct data at the predefined tempo. This was due to hardware and system constraints. With fifteen sensors involved in the system and the hand's rapid motion, the sensory glove was unable to capture the correct orientation at the predefined tempo. A more advanced microcontroller with multiple I2C communication lines could resolve the issue, and further work could be conducted on the structure of the IMU sensor's serial communication network. This was identified as the limiting factor in the present work due to the cost requirements of the sensory glove. However, when the tempo was reduced by a third, the system could capture the motion of the sensory glove.

The final test determined the practicality of the sensory glove in an HRC environment. In order to test this an advanced low-cost manufacturing robot, known as the xArm 5 Lite, was used. The researcher designed and developed an algorithm to simulate an HRC interaction between a worker and a robot. The test was performed to validate the possibility of such a collaboration between the worker and a robot in a safe environment. The efficiency of the collaboration was analysed. The test results showed that the robot was able to collaborate with the worker every time the need for collaboration arose. Multiple trials were performed, and the efficiency of the collaboration was one hundred percent. This demonstrated the potential impact that the sensory glove could make if implemented in an FMS environment. Further research could allow for a greater degree of collaboration with complex tasks that would have the robot and human work hand-in-hand.

While each of the test results showed the overall performance of the sensory gloves, they also demonstrated the effectiveness of such a device in an HRC system. An HRC system can only be established if three key areas, collaboration, coexistence and safety, are achieved. The application test showed that the use of a sensory glove made it possible for such a device to enable a collaborative environment where humans and robots could coexist. For coexistence, the sensory glove needed to have a high level of accuracy and dynamic range to ensure that the worker could collaborate with the robot. The dynamic range and accuracy tests showed that the worker could safely collaborate and coexist in an environment with a robot while they worked together towards a common goal

## 6. Conclusions

Research on current sensory gloves created the foundation for the present study. A literature survey indicated that IMU sensors provided the most effective and cost-efficient solution to tracking a human hand's orientation as discussed in [24]. Through the literature, it was established that the sensory glove should be designed and developed as a mechatronic device. This would ensure that the device encompassed all core components that can create an efficient and cost-effective solution. A detailed design of the core components, which included mechanical, computer, control, electrical and electronic systems, was established for the sensory glove. A high level of integration between the core components was a key criterion in the creation of a mechatronic device.

In order to test and analyze the performance of the sensory glove, three tests were conducted involving three performance metrics. These metrics allowed the researcher to conclude the overall operational performance of the sensory glove. Improvements, such

as using a high-performance IMU known as ADXL 345, and fine-tuning of the control algorithm, could improve the device's performance metrics. The application test focused on validating if the sensory glove could be used in an HRC environment. The sensory glove was integrated into a simulated HRC environment to verify if it could collaborate with a robot. A testing algorithm was built and it showed that the sensory glove was an enabler for collaboration with a one hundred percent efficiency. The results for the three tests also showed that a sensory glove could be used as an enabler for an HRC system as they showed that humans could coexist and collaborate safely in that same working environment. With the total development cost of the sensory glove being 120 euros, it showed that it could be a low-cost solution.

The shortcoming of the current system is that it is a little bit bulky. Future work can focus on making a soft glove with the approach of skin sensors [33]. Other future work could focus on designing a glove with adequate properties of robustness, resilience, and fault tolerance. Differences in these concepts can be found from [34].

The project achieved the objective defined at the beginning of the research. The research, design and results showed that it was possible to develop a low-cost mechatronic sensory glove to enable humans and robots to collaborate in a customized environment in an AMS. The results also showed that safety can be achieved with further improvements in the algorithm design, filtering solution and hardware. This demonstrated the potential impact of the sensory glove if implemented in an FMS environment.

**Author Contributions:** T.B. was the lead author of this paper and conducted the research while under the supervision of S.A. and G.B.; S.A. and G.B. contributed by supervising the research, reviewing the article and editing the article. All authors have read and agreed to the published version of the manuscript.

**Funding:** This research was funded by NRF, grant number: 129048:2020 and also by UKZN research funds.

**Acknowledgments:** I would first like to acknowledge my mother, my father, my grandmother, my fiancé, Miep and Polly for the love and support that they provided me through this research project. A special thanks needs to be given to my supervisors S Adali and G Bright for the opportunity to further my research career in the field of engineering. I would like to thank them for the academic support that they have provided me through this journey. I would like to thank G Loubser, senior electronic technician at the University of KwaZulu-Natal for his guidance and assistance throughout my project.

**Conflicts of Interest:** The authors declare no conflict of interest. The funders had no role in the design of the study; in the collection, analyses, or interpretation of data; in the writing of the manuscript, or in the decision to publish the results.

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
