# Peer review of "Low-Cost Sensory Glove for Human–Robot Collaboration in Advanced Manufacturing Systems†"

_robotics, doi:10.3390/robotics11030056_

Round 1
Reviewer 1 Report
In order to improve the paper, the reviewer suggests the following corrections to the authors:
- Fig. 6 should be redone in accordance with the rest of the figures;
- Rewrite the references [21], [28] do not think that they respect the template of the paper;
- Other comments are in the attached file.

Author Response
To whom it may concern
Based on your feedback, I have made the following changes:
- Updated Figure 6
- Checked format of references 21 & 28
- Used synonyms where possible to reduce repetition
- Corrected grammatical errors that were outlined
- Further explained concepts that were unclear and add additional information where indicated.
- Correct HRC definition
- In terms of the Pearson's Correlation Co-efficient - The raw data values show that there is small error between the measured and simulated values. Pearson's and Bland-Altman show that we can compare these values. This explanation can be found in the text.
- Provided greater detail of procedure of tests in the article.

Reviewer 2 Report
Formal comments:
- The structure of the article is not according to the template.
- Line 33 citation not to be listed individually.
- Figure 6 bad quality.
- Line 407 typo in the text.
- There is a lack of comparison with alternative HRC projects such as Body motion recognition, Voice commands, Haptic interaction instruction and other.
- There is no indication of the main benefits of this project and the reason for its publication.
- This project is nothing special, why doesn't it deal with the concept of multimodal intuitive robot control?
- Why isn't the dynamics of the robot for control with the help of a glove also considered? This would be a problem for larger industrial robots.
- How the author worked to the fact that his glove has a 95% confidence interval ? The use of such cheap components does not give a very high presumption for such accuracy.
- Why didn't the author continue in the many open source projects available that address the given design of somatosensory gloves?
Author Response
To whom it may concern
Based on your feedback, I have made the following changes:
- Checked and corrected the structure for article according to the template.
- Made General corrections
- Updated Figure 6
- Corrected line citation and grammatical errors that were outlined
- Highlight the problems of the researched sensory gloves
- Highlighted the advantages of the proposed design verse researched sensory gloves
- Provided greater detail of procedure of tests in the article.
- Provided greater detail of how the algorithm works in the article.
Reviewer 3 Report
This paper presents a sensory glove for Human-Robot Collaboration (HRC) in a manufacturing environment. The authors developed a low-const sensory glove composed of IMUs and three-dimensional render, and assessed the accuracy, dynamic range and practicality through experiments. They concluded that the proposed glove was an innovative low-cost solution and could provide a safe working environment in HRC.
The introduction is clearly written and the importance of the study is understandable. There are many related studies on a sensory glove as described. Against these backgrounds, this paper also presented a sensory glove with potential. However, the advantages of the proposed grove need to be more well-illustrated by comparing with other related studies. Simulink libraries are helpful for researchers, but detailed explanations of the proposed methodology are necessary for making the contributions clear.
Section 1 and 2:
- The authors present the reviews of the related studies on sensory gloves for HRC in advanced manufacturing. What is the problem of these gloves, which the authors addressed in this paper?
Section 3:
- What does “low-cost” mean? Cheap components? Efficient design process? Low power for operating the developed system? At the end of 3.1, there are some mentions, but they are not clear.
- In 3.2.1, MPU6050 appears the first time and is the product name. Readers would not understand what type of sensor it is and the rest of this paper. After googling it, I found out this is a six-axis gyro and acceleration sensor. Please add the detailed speciation of this sensor.
- The human handle model shown in Fig. 8 looks a rough design of the actual hand, and is inappropriate for industrial applications. For instance, it is difficult to pinch small objects such as screws using the thumb and index finger. Please justify that this model is useful for the applications focused on in this paper.
- Please describe how to obtain the joint angles using equations. There are some paragraphs and Fig 10 in 3.2.3, but they are not clear.
- Please add the description of the applied filters as well. The text in Fig. 12 is not readable because this figure is a too low-resolution image.
- The proposed glove shown in Fig 16 looks similar to the related study in Fig. 2. What is the advantage of the proposed one?
- Since the hand model has 15 joints, I guess, the number of IMU sensors needs to be at least 15 + 1 (for base orientation to measure MP joint angles). However, the glove includes only 15 sensors. How does the system estimate the MP joint angles (Revolute 17, 20, 23, 26 and 29 in Fig. 13)?
- How much does the proposed glove weigh?
- How much is the system sampling frequency? This information is used in the experiment, but there is no exact value of the frequency in the text.
- How long is the time delay? There are several mentions of the time delay, but there is no exact value.
- Please compare these measurements above with the related studies, e.g., devices shown in Figs 2 and 3.
Section 4 and 5:
- What is the unit of values in Table 1 and 2? [Degree]?
- How much is the accuracy and resolution of the goniometer used in the accuracy test?
- The dynamic range would depend on the motion frequency, internal delay and filtering parameters. Please justify that the introduced test is enough for the focused application.
- At the end of the dynamic range test, the authors mentioned “the results showed theta the sensory glove had satisfactory performance in terms of the cost of development.” Is there any relationship between the dynamic range and the development cost?
- How did the system recognize that the grove posture matched the predetermined one in the application test? Please describe the algorithm.
- In the application test, there should be time conditions for an appropriate assessment. How long did it take to unlock the robot arm using the proposed method? How long was the subject able to keep the unlock state? Otherwise, even if the subject unlocked the robot arm after a long trial and continued operating it for a second, the answers to both questions in Table 4 are “Yes,” which does not make sense.
Author Response
To whom it may concern
Based on your feedback, I have made the following changes:
- Highlight the problems of the researched sensory gloves
- Explained what low-cost meant in the article
- Explained what MPU6050 is in the text
- Fixed grammatical errors that were outlined
- Further explained concepts that were unclear and add additional information where indicated.
- Highlighted the advantages of the proposed design.
- Answered the questions in the article.
- Provided greater detail of procedure of tests in the article.
- Provided greater detail of how the algorithm works in the article.
- Time conditions were not considered during the testing process but will be something to consider in future work.
Reviewer 4 Report
Comment 1: In collaborative robotics, the essential problem is to avoid collision between the human worker and the robot. There are other solutions including the skin sensor, see “Tactile and Thermal Sensors Built from Carbon-Polymer Nanocomposites-A Critical Review” (Sensors) , and those that are based on the signal analysis of the torque on the driving axis. The authors need to come late the benefit of the glove solution with respect to the existing solution. Comment 2: It is not clear about the situation the glove sensor is used: teaching or operating? Comment 3: Glove sensor itself is not new in other application situations. So, the authors may want to explain any new stuff with the glove sensor. Comment 4: What is evidence to claim low cost? A comparison with others in terms of cost may be helpful. Comment 5: How is the reliability and resilience of the glove sensor? The definition of resilience can be found from the literature “towards a resilient manufacturing system” (CIRP Annals, 2011). Basically, it refers to the ability of sensor that can still function despite partial damage of the system.Author Response
Reviewer 4
The author added an analysis of existing HRC solutions through camera based, AR and VR technology. The author has added direct benefits of sensory glove solution to existing solutions, such as camera based, AR and VR technologies that are currently being used in HRC systems. This can be found in the beginning of the Introduction section of the article.
The author has made it more clear, throughout the article, that the collaboration is in the form of operation of the robot.
The authors main focus is presenting a sensory glove solution for an HRC environment through operation of a robot. The development of new technology for a sensory glove was not the focus of the research paper.
The author has added a cost analysis of existing sensory gloves in comparison. This can be found in the discussion section.
The paper did not consider the resilience of the glove if it withstood damage but will be considered in future research.
Reviewer 5 Report
The paper is e revised version of a previous submission, and it looks quite good and almost ready for publication
I would suggest to delete figure 1, which is very well known and not useful in a scientific paper, and to edit figure 6, which looks like being partially hand written.
The multibody model of figure 9 does not add information to the paper, which is rather long. I would suggest to delete this part.
Author Response
Figure 1 - It shows the reader how a graphical representation of Mechatronics as a subject and how it relates to the project.
Figure 6 - It was drawn on AUTOCAD, which is a drawing software. I have re uploaded the image into the manuscript and hopefully that will improve the quality
Figure 9 - The simulink system is described above and the figure gives the reader a visual representation of the system. It shows how it was converted from a CAD object to a system of joints and links.
Reviewer 6 Report
This paper presents an approach in the human-robot collaboration using sensory gloves. Overall, this paper proposed an interesting approach with low-cost sensors in the human-robot collaboration. However, the level of human-robot collaboration is not really clear described in this paper. This paper dominantly describes how they build the sensor. Also, to enhance the quality the author can improve the background for this research. Some suggestions are presented for clarifying some points in each section that are summarized as follows:
Section 1
- The introduction section needs more enhance to describe the background in this research
- There is needed to build a good sequence of telling the problem, urgency, research gap, and the novelty of the proposed approach
- There are two introductions in sections 1 and 2, what is the difference?
- Some paragraphs only consist of one sentence, the author should write more than 4 sentences in the paragraph
- Many citation sentences only inform the references considered in this study, however, this paper fails to describe what the author considers from this research in building their research
- Why does the author propose a low-cost sensor? It is needed to inform before the objective
- A lot of approaches can consider in building human-robot collaboration like vision system or physiological response such as EMG, the author has to inform the urgency use sensory gloves.
- Why the author considers the human finger as information not consider another human hand like wrist or carpal to limit the use of sensor
Section 2
- This section did not need an introduction
- For the end of sections, the author should conclude with explanation the use of previous explanations in their research
- The author should provide the proper image (difficult to read the written text in the figure)
Section 3
- This section did not need an introduction
- Figure 7 need the proper image to explain the PCB board
- Authors should add citations if they refer Arduino to develop the design of PCB
- Could authors inform about the resolution of the data?
- The author should provide the proper image (difficult to read the written text in the figure)
- Why did the author add a conclusion for this section?
- Is this sensor directly gather the digital data? Did the author convert the analog to digital?
Section 4
- The author needs to illustrate the testing with the figure because it is difficult to imagine the case from the description
Section 5
- The author provides the flowchart with proper text (difficult to read)
- I didn’t see the collaboration scenario between humans and robots using sensory gloves, the author should inform proper scenarios in human-robot collaboration, or try to change the collaboration type, like human control the robot or etc.
Author Response
As indicated, The author has expanded the background of the research. This was done by adding an analysis of existing HRC solutions through camera based, AR and VR technology. The author added direct benefits of sensory glove solution to existing solutions, such as camera based, AR and VR technologies that are currently being used in HRC systems. This can be found in the beginning of the Introduction section of the article.
The introduction also further elaborates on the problem, research gap and the novelty of the proposed approach. The researcher also improves the paragraphing in the article.
The introduction in section 2 details the focus areas of the literature.
The author has explained what he considers in his research in building the research. This is present in the introduction and literature review when the researcher considered existing solutions for HRC environments.
The Author highlights that sensory gloves that are currently available and at substantial cost. Therefore to make it more affordable, a low cost solution is developed.
Other hand parts, such as wrist, add greater dimensions and complexity to the problem. Therefore, the project focuses on just the finger to establish a prototype for such application
The author has improved the resolution of the images in the article.
The MPU605 has a built in Analogy to Digital converter so we receive Digital data.
The author has introductions in Sections 2 and 3 to outline the chapter.
The conclusion that sits in Section 3 summarises the design and development of the sensory glove.
The collaboration scenarios is present in the Application test. The flow chart details the process and the images provide clarity on how it works.
Round 2
Reviewer 1 Report
I suggest the authors to complete the paper with Fig. 6 redrawded.
Author Response
Thank you for your comments.
Reviewer 3 Report
My major concern is that there are no equation-based explicit explanations of the proposed methodology, which is necessary for a robotics journal paper. Again, I understand that Simulink is helpful for researchers to try rapid development. However, the employed algorithm should be explicitly described in an academic report. The following comments are additional my questions:
Section 1:
- There are many many other sensor gloves based on IMUs. Please add these references and explain the novelty of your proposed glove.
For example: Lin BS, Lee IJ, Yang SY, Lo YC, Lee J, Chen JL. Design of an Inertial-Sensor-Based Data Glove for Hand Function Evaluation. Sensors (Basel). 2018;18(5):1545. Published 2018 May 13. doi:10.3390/s18051545 - The authors pointed out the computational power problem of the related grove in [21] toward a wireless system. However, the proposed system in this paper is also not wireless. Is this compatible with wireless communication?
- The authors pointed out the component cost (price) in the reviews of the related studies. However, the proposed system is developed using Simulink, which is an expensive commercial product. Why did you not develop with other free programming languages?
Section 3:
- The human handle model shown in Fig. 8 looks a rough design of the actual hand, and is inappropriate for industrial applications. For instance, it is difficult to pinch small objects such as screws using the thumb and index finger. Please justify that this model is useful for the applications focused on in this paper.
- Which sensor is the palm sensor that appeared in L.476 corresponding to in Fig. 16? Each finger has three joints, and the number of the installed sensors is 15. I could not find any information on this palm sensor in the figures.
- Since MPU6050 does not include a magnetometer, yaw drift would not be able to be canceled out. Does this drift affect the system's accuracy?
- How did you calculate 15 joint angles from the 3D orientation data measured by 15 sensors?
- Please add the description of the applied filters, including filtering parameters.
- How much is the system sampling frequency? This information is used in the experiment, but there is no exact value of the frequency in the text.
- How did you measure the time delay (80 ms)?
Section 4 and 5:
- If you consider an industrial application, I think spatial position accuracy is more important than joint angle accuracy. However, there is the only discussion on the joint angles. How much is the spatial accuracy? This would depend on the hand model and the glove design as well as the sensor processing.
- From an academic viewpoint, I could not understand the sentence L. 665 "There was a high level of accuracy due to an average of 10 percent difference between the simulated and measured value." If you evaluate something, you need to know the tool's accuracy or confirm its accuracy with another known tool. In Table 1, all data values from the goniometer are ended with “.0”, Is this correct from a viewpoint of significant figures?
- All joint angles never match the predetermined angles simultaneously. How did the system recognize that the grove posture matched the predetermined one in the application test?
- If you did not consider any time conditions in the last experiment, please justify the result of the experiment as a scientific report. Otherwise, I cannot find the scientific (academic) meaning of this result.
Author Response
The research paper's goal was to outline the potential prototype of such a device in a Human Robot Collaborative system. Extensive research has been done on the design on sensory gloves for other fields of research but little has been done on HRC systems. The article presents a novel solution to HRC system that has the potential to create solution, especially at a low cost.
The researcher has added a cost analysis of existing sensory gloves in comparison to the developed one by the researcher. This can be found in the discussion section.
The author added an analysis of existing HRC solutions through camera based, AR and VR technology. The author has added direct benefits of sensory glove solution to existing solutions, such as camera based, AR and VR technologies that are currently being used in HRC systems. This can be found in the beginning of the Introduction section of the article. This highlights the novelty of the solution.
The system is compatible to a wireless system
The research has extensive experience with Simulink and found it to be the best solution at the time. This consideration will be taken into future studies.
The hand model was based on the ideology that the user would do simple tasks while in collaboration with the robot. The use of small objects, like screws , would not be used by the worker due to their specific environment on the manufacturing floor.
There was no sensor that was placed on the palm and there is no sensor on Fig 16.
As the MPU6050 did not include a magnetometer, the data values were passed through an averaging filter as well as the cleaning function to reduce noise in the sensors.
The time delay of the sensory glove was determined by the initial testing that was done on the simulink software.
The spacial accuracy of the glove was not tested. However, it is something that we note for further research.
With regards to significant figures of the table, you are correct.
The system was given a range of values to accept when determining the glove orientation position.
The scientific report refers to the an experimental investigation of such device being used on a HRC system. With the results presented, it showed the potential of such device to be used to effectively collaborate with a robot.
Reviewer 4 Report
The authors need to include a discussion of future work, as they agreed. The current system is bulky in my opinion. I suggest the following paragraph: “The shortcoming of the current system is a little bit bulky. The future effort may be taken on making a soft glove with the approach of the skin sensor [a]. Another future work is on designing a glove with an adequate property of robustness, resilience, and fault-tolerance. Differences of these concepts can be found from [b].”
[a] “Tactile and Thermal Sensors Built from Carbon-Polymer Nanocomposites-A Critical Review” (Sensors). doi: 10.3390/s21041234
[b] On the principle of design of resilient systems – application to enterprise information systems, Enterprise Information Systems, 4:2, 99-110, DOI: 10.1080/17517571003763380.
Author Response
Hi
Thanks so much for the feedback, I have added the section in, in my conclusion
Reviewer 6 Report
Some suggestions are already satisfied. However, still, some parts need to be improved:
Section 1
- I think the author could discuss camera-based for human-robot collaboration than using AR or VR as background. This is because camera-based human-robot collaboration is the same in the real environment.
- The author cites some research at lines 54-58, however, they do not tell about the insight for this research at end of paragraph
Section 2
- Still same this section did not need an introduction as section title (Just remove introduction title)
Section 3
- Still same this section did not need an introduction (Just remove introduction title)
- Could authors inform about the resolution of the data?
- I cannot find figure 9
- Still for several images like figure 12, 13, and 14 difficult to understand because the text is too small
Author Response
Hi
I do briefly touch on camera based solution in the Introduction. I believe that AR and VR research are relavent as they are all examples of existing solutions in HRC systems.
I have removed the introduction sections and fixed the figure problem.